# Cachexia: Pathophysiology and Ghrelin Liposomes for Nose-to-Brain Delivery

**DOI:** 10.3390/ijms21175974

**Published:** 2020-08-19

**Authors:** Cecilia T. de Barros, Alessandra C. Rios, Thaís F. R. Alves, Fernando Batain, Kessi M. M. Crescencio, Laura J. Lopes, Aleksandra Zielińska, Patricia Severino, Priscila G. Mazzola, Eliana B. Souto, Marco V. Chaud

**Affiliations:** 1Laboratory of Biomaterials and Nanotechnology (LaBNUS), University of Sorocaba, Sorocaba, 18078-005 São Paulo, Brazil; cecilia.barros@edu.uniso.br (C.T.d.B.); alessandra.rios@edu.uniso.br (A.C.R.); thaisfrancine1@hotmail.com (T.F.R.A.); fbatain@gmail.com (F.B.); kessicrescencio@yahoo.com.br (K.M.M.C.); laurajulialopes@gmail.com (L.J.L.); 2Department of Pharmaceutical Technology, Faculty of Pharmacy, University of Coimbra, Pólo das Ciências da Saúde, Azinhaga de Santa Comba, 3000-548 Coimbra, Portugal; zielinska-aleksandra@wp.pl (A.Z.); souto.eliana@gmail.com (E.B.S.); 3Institute of Human Genetics, Polish Academy of Sciences, Strzeszyńska 32, 60-479 Poznań, Poland; 4Institute of Technology and Research, University of Tiradentes (UNIT), 49032-490 Aracaju, Sergipe, Brazil; pattypharma@gmail.com; 5Tiradentes Institute, 150 Mt Vernon St, Dorchester, MA 02125, USA; 6Center for Biomedical Engineering, Department of Medicine, Brigham and Women’s Hospital, Harvard Medical School, 65 Landsdowne Street, Cambridge, MA 02139, USA; 7Faculty of Pharmaceutical Science, University of Campinas (UNICAMP), Candido Portinari Street, Campinas, 13083-871 São Paulo, Brazil; pmazzola@fcf.unicamp.br; 8CEB—Centre of Biological Engineering, University of Minho, Campus de Gualtar, 4710-057 Braga, Portugal; 9Bioprocess and Biotechnology College, University of Sorocaba, Sorocaba, 18078-005 São Paulo, Brazil

**Keywords:** cachexia, ghrelin, liposomes, nose-to-brain

## Abstract

Cachexia, a severe multifactorial condition that is underestimated and unrecognized in patients, is characterized by continuous muscle mass loss that leads to progressive functional impairment, while nutritional support cannot completely reverse this clinical condition. There is a strong need for more effective and targeted therapies for cachexia patients. There is a need for drugs that act on cachexia as a distinct and treatable condition to prevent or reverse excess catabolism and inflammation. Due to ghrelin properties, it has been studied in the cachexia and other treatments in a growing number of works. However, in the body, exogenous ghrelin is subject to very rapid degradation. In this context, the intranasal release of ghrelin-loaded liposomes to cross the blood-brain barrier and the release of the drug into the central nervous system may be a promising alternative to improve its bioavailability. The administration of nose-to-brain liposomes for the management of cachexia was addressed only in a limited number of published works. This review focuses on the discussion of the pathophysiology of cachexia, synthesis and physiological effects of ghrelin and the potential treatment of the diseased using ghrelin-loaded liposomes through the nose-to-brain route.

## 1. Introduction

Cachexia is a multifactorial syndrome characterized by the continuous loss of skeletal muscle mass (with or without loss of fat mass) that cannot be entirely reversed by conventional nutritional support, leading to progressive functional impairment [1,2]. Cachexia is described in association with many chronic conditions, infectious diseases and seen in patients after extensive traumatic injuries or sepsis [3,4]. Until now, no approved drug intervention has effectively and completely reversed the findings of cachectic syndrome [5]. Several promising treatment approaches have failed to meet the challenge of phase III clinical trials. Additional advances are urgently needed [6]. Studies have shown positive and encouraging effects of ghrelin in the treatment of cachectic patients [7,8,9]. While the data is promising to support the therapeutic use of ghrelin in cachexia, treatment drawbacks that limit its clinical use include its short half-life and the need for parenteral administration [10,11]. In this context, the intranasal release of ghrelin formulated in liposomes able to cross the blood-brain barrier (BBB) or a central action has been proposed as a promising alternative to improve the bioavailability of this drug [12]. This process is schematically represented in Figure 1. The disadvantage of the therapeutic use of the peptides is their susceptibility to cleavage by enzymes [13]. Consequently, strategies using drug delivery systems, such as liposomes, can be developed to protect the peptides against physiological instability, enzymatic attack and improve their permeation [14].

The nasal route has been, successfully, exploited for the continuous release of drugs, including macromolecules, to the central nervous system (CNS) [15]. The olfactory region of nasal mucosa provides a connection between the nose and the brain that can be used for a more easy distribution of drugs that act in the CNS [16]. The nose-to-brain delivery of ghrelin-loaded liposomes may be an approach to protect ghrelin against biodegradation and improve the accessibility of the ghrelin to its brain targets. Therefore, liposomes may be able to protect ghrelin from nasal enzymes, improve permeation and, consequently, increase ghrelin bioavailability, which can be promising in the treatment of cachexia.

## 2. Cachexia

Etymologically, cachexia is derived from the Greek *kakos* (bad) and *hexis* (condition). In Greece, in the fourth century BC, Hippocrates accurately described the central pathogenesis of cachexia, saying: “meat is consumed and becomes water”, considering cachexia as a sign of death [17,18]. Cachexia has a multifactorial and systemic character, which compromises the definition of the criteria for its diagnosis [2,19]. Weight loss and inflammation are nevertheless common in all clinical guidelines [1], together with metabolic pathway changes in many tissues and organs [20]. Cachectic patients, in addition to functional impairment, have a compromised quality of life, increased mortality and greater susceptibility to toxicities related to the treatment of the associated disease, leading to a reserved prognosis [2].

The disease is characterized by the continuous loss of skeletal muscle mass, fat and bone [21]. However, the loss of muscle tissue is considered the main pathophysiological mechanism to explain the reduction of the physical capacity, increased fragility, susceptibility to disease progression, increased hospitalization rate and, consequently, increased mortality [22].

Cachexia is a significant and growing public health problem [23]. It is a serious condition but underestimated and not recognized in patients. Clinicians and researchers focus their efforts and attention on the primary disease, rarely recognizing cachexia as a distinct condition, in addition to the lack of effective therapy that would justify the recognition and registration of cachectic conditions [23].

Cachexia is described in association with many chronic conditions (Figure 2), such as cancer, chronic heart failure (CHF), rheumatoid arthritis, chronic obstructive pulmonary disease (COPD), chronic kidney disease, liver cirrhosis, cystic fibrosis, Crohn’s disease, stroke and degenerative neurological disorders [24,25]. It has also been seen in patients after traumatic injuries and extensive sepsis, in addition to being associated with patients with infectious diseases such as HIV/AIDS, tuberculosis and malaria [3]. A hypothesis has been proposed that, regardless of the specific chronic disease, the loss process follows a typical final metabolic pattern. This metabolic pattern is usually related to an advanced stage of the underlying disease and can best be summarized as an increase in catabolic turnover and anabolic weakness [22]. Regardless of the underlying disease, cachexia is associated with an inadequate response to drug treatment, poor quality of life, poor prognosis and increased mortality compared to non-cachectic patients [22].

Data from 2014 in industrialized countries, such as countries in North America, Europe and Japan, showed that the general prevalence of cachexia is growing and has reached about nine million patients [6]. Data are scarce for countries in South America and Africa; it is estimated that cachexia is also a major problem in these countries.

Cachexia can be qualified as a global public health problem. For a person to be classified as such, comorbidity must: (i) have a high burden in terms of morbidity, mortality, quality of life and costs; (ii) be unfairly distributed, affecting disadvantaged population groups to a greater extent; (iii) there is some evidence that public health strategies can substantially reduce the burden of the disease and, finally, that these preventive strategies are not yet fully installed [4]. Besides, these, there is a significant increase in life expectancy worldwide, resulting in a higher percentage of older adult people. Consequently, there is clear evidence of an epidemiological transition, resulting in a marked increase in the incidence of chronic diseases worldwide [26]. Low-income countries still face a double-burden of disease as they continue to tackle infectious disease problems. There should be an increase in awareness of cachexia and help in the community to understand its complexity and magnitude [4].

Adequate nutritional support remains one of the pillars of the treatment of cachexia [5]; however, the loss of body mass associated with cachexia is not only mediated by decreased food intake [27]. Thus, patients on total parenteral nutrition and, therefore, with perfectly controlled energy intake still lose weight and suffer from symptoms of cachexia [20].

The additional weight loss is due to processes related to metabolic changes, mediated by the excessive release of proinflammatory cytokines and the increased activity of the sympathetic nervous system. Both catecholamines and proinflammatory cytokines promote catabolic processes [28].

Proinflammatory cytokines decrease the effectiveness of growth hormones (GH) and act on the central nervous system as mediators of inflammation and as catabolic factors that stimulate proteolytic pathways, leading to muscle atrophy and the increased breakdown of adipose tissue [27].

Proinflammatory cytokines, including interleukin-6 (IL-6), IL-10 and IL-1β and tumor necrosis factor-alpha (TNF-alpha), stimulate the breakdown of muscle proteins, cause contractile dysfunction and inhibit myogenesis, in addition to promoting the waste of adipose tissue, the inhibition of adipocyte differentiation, stimulation of lipolysis and increased apoptosis in adipocytes [28].

The negative energy balance related to cachexia is associated with the inflammatory signaling in the hypothalamic melanocortin system; the regulation of the central inflammatory signaling of the metabolism is related to melanocortin signaling. Neurons that express pro-opiomelanocortin peptide (POMC), orexigenic peptide associated with agouti (AgRP) and neuropeptide Y (NPY) in the presence of proinflammatory cytokines generate a decrease in the release of AgRP/NPY and increase the release of POMC [29].

POMC is cleaved to produce melanocyte-stimulating hormone (alpha-MSH). Alpha-MSH, released at synapses, is linked to melanocortin 4 (MC4R) receptors, leading to an increase in the basal metabolic rate, a reduction in lean body mass and a decrease in food-seeking behavior. AgRP/NYP, a natural MC4R agonist, when decreased, increases the effects of POMC, leading to reduced appetite, resulting in appetite restriction generated by the decreased expression of AgRP/NYP (a natural MC4R agonist), associated with an increase in the effects of POMC [27].

The TNF-alpha content also decreases carbohydrate reserves, inhibiting insulin receptor activity and a decrease in the expression of the glut-4 transporter, which should play a key role in impaired glycemic homeostasis [30].

The increased activity of the sympathetic nervous system resulting from increased plasma concentrations of neurotransmitters also results in a loss of body mass and increased energy expenditure, stimulation of lipolysis, decrease in lipogenic enzymes and stimulation of apoptosis in skeletal muscles [28]. Catecholamines have been linked to an enhanced immune response, suggesting that the sympathetic nervous system is an essential mediator of cachexia [27].

When starting an inflammatory process, the body adopts an increase in the systemic activity of the sympathetic nervous system. One of the goals of inflammation is to stimulate dendritic cells, which participate in the immune response [31]. Immune cells express receptors for neurotransmitters that are functional and translate neuronal signals into signals from immune cells [32]. However, if this configuration of increased sympathetic nervous system activity persists, the effects are detrimental due to the resulting chronic catabolic state, leading to cachexia, high blood pressure, insulin resistance and increased cardiovascular mortality [31].

As cachexia is associated with complex pathophysiological processes, pharmacological treatments, such as calcium supplementation or appetite stimulants, such as medroxyprogesterone acetate; megestrol acetate; cyproheptadine and corticosteroids, such as prednisolone, methylprednisolone and dexamethasone, are used in the current treatment and present only limited success [33]. The pharmacological treatments available do not comprehensively address the relevant components of cachexia syndrome [7].

There is an increase in observations and discussions by scientists about cachexia being seen as a common final metabolic pathway, regardless of the underlying disease being a distinct and treatable condition [22]. Research on cachexia is still underdeveloped, but signs can already to be seen that indicate an essential scientific effort that will evolve into clinical studies, generating the hope that effective therapies for this syndrome will be developed in the coming years [23].

## 3. Ghrelin

The administration of ghrelin in patients with cachexia results in the decreased release of proinflammatory cytokines and reduces sympathetic nerve activity [34]. The use of ghrelin is encouraged in the treatment of cachexia by the potential to stimulate anabolic activity, promote growth hormone secretions, regulate the autonomic nervous system and suppress the effects of inflammation [35].

Ghrelin is a 28-amino acid hormone (Figure 3) produced primarily in the oxyntic mucosa of the stomach [36]. This hormone binds to the type 1a growth hormone secretagogue receptor (GHS-R1a) in the hypothalamus and stimulates the release of growth hormones [37]. Acylation catalyzed by ghrelin-O-acyltransferase (GOAT) is essential for the binding of ghrelin to the receptor (GHS-R1a) [37].

Acylation may not be necessary for all actions of ghrelin [36]. Some evidence shows that the biological functions of deacylated ghrelin are independent of GHS-R1a. Deacylated ghrelin appears to have its cognate receptor. However, this receiver remains undetermined [35].

In the acylation of ghrelin, GOAT uses dietary triglycerides, including C6-C10 fatty acids, with a strong preference for C8 [38]. The ghrelin-GOAT system has functions in the regulation of energy homeostasis, including the ability to communicate the current peripheral nutritional status to the hypothalamus and perform energy compensations [39].

GOAT seems to act also to detect and communicate to the brain about the availability of peripheral nutrients and, also, for energy storage [35]. GOAT is necessary for the prevention of hypoglycemia under conditions of starvation by maintaining blood glucose levels mediated by growth hormones [40].

The discovery of ghrelin represents an important turning point in the study of stomach-brain interactions and has made enormous contributions to our understanding of systemic homeostasis [35].

One of the mechanisms of action proposed for ghrelin assumes that this hormone regulates the metabolism through the activation of orexigenic neural circuits, such as the central melanocortin system [39]. In addition, to growth hormone secretions, the stimulation of ghrelin in the hypothalamus results in a decrease in the expression of the effects of the anorexigenic POMC peptide and an increase in the expression and release of the orexigenic peptides AgRP and NPY [27,41].

Additionally, to decrease in the expression of POMC expression, inflammatory IL-1 beta signaling in the melanocortin system is strongly associated with the negative energy balance related to cachexia. Ghrelin has decreased POMC expression and inhibits proinflammatory cytokines, such as IL-1alpha, IL-1beta and TNF-alpha [29,42].

Most animals, including humans, have standardized eating patterns in which meals are eaten based on learned and/or environmental factors. The signaling of ghrelin with ventral hippocampal neurons (vHP) is physiologically relevant for conditioned feeding behaviors. The activation of neurons by ghrelin-vHP still communicates directly with neurons in the lateral hypothalamus (LHA) expressing neuropeptide orexin [43]. Together, the central signaling of ghrelin-GHS-R1a induces feed directly through the activation of NPY/AgRP neurons and indirectly through the vHP-LHA pathway [35].

The synthesis of ghrelin and routes involve ghrelin signals in the hypothalamus. In X/A-like cells located in the stomach, it is where the pro-ghrelin is acylated. Two routes were proposed to transmit the signals of ghrelin derived from the stomach to the hypothalamus: the afferent vagal nerve and the blood circulation. The vagus nerve is the tenth cranial nerve, contains both efferent and afferent fibers and transmits information from the viscera to the brain. Ghrelin binds to GHS-R1a and suppresses the electrical activity of the vagal afferent nerve. This electrical signal reaches the NTS, (nucleus tractus solitarius) which synapses with NPY (neuropeptide Y) neurons in the ARC. (arcuate nucleus of the hypothalamus). The circulating ghrelin is transported through the BBB and binds to neurons in the vicinity of capillaries [35].

GHS-R1a in the hypothalamus is predominantly expressed in ARC. ARC contains orexigenic neurons expressing NPY and AgRP and anorexigenic neurons expressing POMC. Both NPY/AgRP neurons and POMC neurons project to vHP. The high activity of POMC neurons increases the release of alpha-MSH in vHP, which, in turn, acts on neurons expressing MC4R to suppress the food intake. NPY acts by stimulating the food intake, while AgRP antagonizes MC4R. The action of ghrelin on vHP stimulates feeding also with the activation of the neurons LHA (lateral hypothalamic area) and orexin release. Ghrelin induces the food intake by activating the NPY/AgRP neurons. Ghrelin stimulates the release of GH by activating the somatotrophs in the anterior pituitary [35].

Ghrelin also suppresses systemic inflammation through its sympatho-inhibitory functions. Central ghrelin receptors that involve an NPY receptor-dependent pathway mediate ghrelin inhibitory properties on norepinephrine release. The modulation of overstimulated sympathetic nerve activation may result in the inhibitory effect of ghrelin on TNF-alpha production [44]. In sympathocytes, the release of norepinephrine by the postganglionic sympathetic nerves increases the output of TNF-alpha [45], and the peripheral administration of ghrelin decreases the circulating levels of TNF-alpha and norepinephrine [44]. Ghrelin also decreases the release of proinflammatory cytokines by the activation of the vagus nerve [35].

GHS-R1a is also expressed in the vagus nerve [46]; the vagus nerve serves as a channel for neurenteric communication, where the increased activity of the vagus nerve, both central and peripheral, leads to increased gastrointestinal motility, increased exocrine pancreatic function and changes in neuroendocrine profiles [47]. The vagus nerve also appears to play a central role in inhibiting the release of proinflammatory cytokines, and studies suggest that this anti-inflammatory activity of vagal stimulation is mediated by ghrelin [48,49,50,51].

Thus, the action of ghrelin depends on the accessibility of the hormone to its cerebral targets [52] but, also, performs actions pertaining to cachexia that are not limited to these central effects. In addition to the central biological activity, ghrelin protects critical organs from the metabolism of stress and metabolic inflammation [35].

In inflammation, GHS-R1a is expressed on lymphocytes, and ghrelin has been shown to decrease the expression of inflammatory cytokines in monocytes and T cells [42] and suggests that ghrelin gene products may play a role in both acute and chronic inflammatory states. This observation further supports this hypothesis that circulating levels of ghrelin are often altered in inflammatory states [50].

In the cardiovascular system, GHS-R1a is expressed in the heart and aorta. It is also reported that the GHS-R1a gene can be detected in the cardiomyocyte cell line in culture and human vascular endothelial cells [34]. Ghrelin presents actions in the cardiovascular system and identify the dilatation of the arterial caliber independently of the endothelium, the decrease of the average arterial pressure and the neutralization of the renin-angiotensin system [53] and suppress the cardiac sympathetic activity. When ghrelin is administered chronically in the case of heart failure, it promotes a reduction in cardiac remodeling after ischemia, improvement in the left systolic function and decreases mortality from fatal arrhythmias [34].

Ghrelin exhibits gastroprokinetic activity [51], because, in addition to increasing vagus nerve activity, it has a structural similarity with motilin and accelerates the rate of gastric emptying even in the presence of vagotomy [54,55]. It is capable of stimulating acid secretion in association with gastrin; however, only when ghrelin is administered centrally in rats, it stimulates colonic motility [56]. Ghrelin is also involved in the regulation of glucose metabolism [57]. Acylated ghrelin mainly has hyperglycemic effects and promotes insulin resistance [58], acting on pancreatic islets to suppress insulin secretion [59] and, in hepatocytes, stimulate glucose production, whereas deacylated ghrelin can neutralize hyperglycemic effects. Deacylated ghrelin increases insulin sensitivity [60]. Consequently, there has been some interest in the potential of ghrelin antagonism to improve diabetes and hyperglycemia [61]. Insulin is associated with the inhibition of ghrelin release, so low insulin levels lead to increased ghrelin secretions [62].

The administration of ghrelin in rats has been reported to prevent muscle atrophy by increasing the phosphorylation of protein kinase B (a protein kinase that plays a key role in apoptosis, cell proliferation and cell migration); decrease the myostatin pathway (protein involved in inhibiting the growth and regeneration of skeletal muscles); activate myogenin (a transcription factor involved in myogenesis and repair) and activate myoD (a protein that plays a vital role in the regulation of muscle differentiation) [9].

In adipose tissue, ghrelin stimulates the expression of genes encoding fat storage-promoting enzymes such as lipoprotein lipase (LPL), fatty acid synthase (FAS), acetyl-CoA carboxylase α (ACCα) and stearoyl-CoA desaturase-1 (SCD1), which can provide important reserves of energy for the organism. In brown adipocytes, ghrelin decreases the expression of thermogenic-related proteins, thereby decreasing the metabolic process during which the body burns calories to produce heat [28].

Due to the benefits, the ghrelin pharmacological approach is considered a promising and valuable approach for the treatment of a variety of metabolic complications, including cachexia [61,63].

Treatment with ghrelin has drawbacks, which include a short half-life and the need for infusion or in bolus parenteral administration, with multiple side effect events like somnolence, a warm feeling, facial warmth, abdominal pain emesis and vertigo [10,11].

The limitations of treatments with exogenous ghrelin have led to the investigation of alternatives such as anamorelin, a ghrelin agonist that is potent and highly specific [64,65,66], and the oral administration of rikkunshito, a ghrelin-enhancing herbal medicine that increases plasma levels of acyl ghrelin [67] and investigation strategies to avoid degradation and improve ghrelin accessibility to targets, which is the focus of this work.

## 4. Nose-to-Brain Delivery

As stated above, the action of ghrelin depends on the accessibility of the hormone to its cerebral targets. However, the actions of ghrelin pertinent to cachexia are not limited to these central effects [35,52]. Thus, the intranasal release of ghrelin for transposition of the BBB. Allowing central action and permeation into the systemic circulation through the highly vascularized nasal mucosa may be a promising alternative.

The brain is isolated and protected by various mechanisms from the external environment. The physiological barrier has properties such as the control of influx and efflux transporters, expression of narrow junctions or by the metabolizing enzyme present in endothelial cells [68,69,70]. The BBB and the cerebrospinal fluid barrier (CSFB) represent the main boundaries between peripheral circulation and the central nervous system [71].

Nose-to-brain may be advantageously used for the chronic brain administration of large and sensitive compounds, such as biotherapeutics [12]. The nose-to-brain delivery system is emerging as a promising approach to the delivery of drugs that require action on the central nervous system.

Intranasal administration is an alternative to oral and intravenous routes and is useful in systemic drug administration and a potential alternative for invasive methods to overcome the BBB/blood-cerebrospinal fluid barrier and deliver drugs to the central nervous system [72].

Current formulations for ghrelin are being planned for parenteral administration [73]. However, when chronic administration is required, invasive injectable routes can lead to poor patient compliance and subsequent treatment failure [74]. The intravenous route still exposes ghrelin to plasma enzymatic degradation [11]. For systemic delivery, the nasal mucosa has the advantage of being richly supplied with blood and has a large surface area, making it an ideal place for drug absorption. Blood flow is essential to remain in the concentration gradient of the absorption site for the blood. In the nose-to-brain route, ghrelin can permeate into the systemic circulation through the nasal mucosa, which is highly vascularized. As in intravenous administration, it may or may not cross the BBB and enter the brain but may also be directed to the central nervous system by the epithelium olfactory or trigeminal nerves.

Drugs delivered intranasally are transported along olfactory sensory neurons to produce significant concentrations in the CSF and olfactory bulb. The olfactory region of the nasal mucosa that provides a connection between the nose and the brain is used to target drug molecules that act on the central nervous system [16].

Then, the mechanisms of nose-to-brain drug transport were described: (i) the systemic route is characterized by absorption of the drug through the nasal mucosa, systemic distribution and passage through the BBB, similar to the intravenous route; (ii) the olfactory pathway in which the drug may permeate the olfactory epithelium to the olfactory bulb from which it has access to the central nervous system; (iii) the trigeminal pathway, the pathway of most significant interest in our study, in which the drug avoids the BBB, is carried by the trigeminal nerve pathway [75].

The systemic pathway is primarily responsible for the transcellular release of low molecular weight lipophilic substances, which can be absorbed more readily into the bloodstream, exhibiting a profile similar to that of an intravenous injection [76]. The systemic pathway is also associated with the hepatic and renal metabolisms of drugs, which can generate systemic exposure without specificity for brain tissues [77].

The olfactory pathway can be subdivided into two pathways: neural and epithelial [78,79,80]. In the neural pathway, olfactory neurons in the epithelium capture xenobiotics through endocytosis, which thus reach the olfactory bulb through the axonal transport of olfactory neurons. In the epithelial pathway, xenobiotics traverse the spaces between the olfactory neurons in the olfactory epithelium and are transported to the olfactory bulb. After reaching the olfactory bulb, xenobiotics can enter other brain regions by diffusion [81]. Only substances of similar size or smaller than the diameter of human olfactory axons (100-700 nm) [82] can be transferred via this route [83].

The trigeminal nerve is the major cranial nerve; although the trigeminal nerve endings are not directly exposed in the nasal cavity, it is assumed that the initial entry point is probably the ophthalmic and maxillary branches of the trigeminal nerve, which innervate the dorsal nasal mucosa together with the anterior part of the nasal cavity and the lateral walls of the nasal mucosa [77,84]. Numerous factors influence the release of drugs to the central nervous system and may determine which of the above pathways may predominate in terms of the extent of drug absorption. Three routes may contribute independently or synergistically to the transport of drugs and for the affinity of a treatment to a particular pathway, which can be modulated by itself or by the formulation properties increasing the permeability and decreasing the mucociliary clearance. Between the strategies possible for increasing the retention time, system biomimetic nanoparticulate modulated with an electropositive surface on lipid-based nanostructures is a highly promising approach.

Nose-to-brain delivery suffers from limitations, such as mucociliary clearance, enzymatic metabolism and permeation limited by particle size. Therefore, liposomes may be able to protect ghrelin from nasal enzymes, and liposome functionalization may increase the permanence time in the nasal cavity, improving permeation and, consequently, increasing the bioavailability of ghrelin, which may be promising in the treatment of cachexia [12].

Nasal drugs circumvent the gastrointestinal and hepatic first-pass effect. However, they may be significantly metabolized in the lumen of the nasal cavity or during passage through the nasal epithelial barrier, due to the presence of a wide range of metabolic enzymes in the nasal tissues. Even though the first-pass nasal metabolism is weaker than the hepatic and intestinal metabolisms, it has a significant importance in drug bioavailability.

The mucociliary clearance system plays an essential role in the defense of the respiratory tract. Mucus collects foreign particles and lashes and provides the driving force by preventing airborne xenobiotics from being inhaled. Xenobiotics adhere to mucus and are transported to the nasopharynx. This mucociliary clearance significantly influences nasal drug absorption [85].

To overcome these difficulties, from nasal metabolism and mucociliary clearance, strategies such as liposome loading can be followed. The use of functionalized liposomes may promote prolonged contact between the drug and the site of absorption, facilitate direct absorption through the nasal mucosa and protect against enzymatic metabolism.

## 5. Liposomes

Liposomes are self-organizing vesicles of nanometer range consisting of concentric lipid bilayers with an internal aqueous phase [86,87]. Many liposomes may be produced with distinct characteristics, which depend on the nature of the lipid components, their surface charge and their possible chemical modifications [88].

The main composition of the liposomes is the phospholipids, which are amphiphilic molecules formed by a hydrophilic head and hydrophobic chains. By these characteristics, when the phospholipids are dispersed in aqueous solutions, they tend to form membranes [89]. Their polar heads prefer to interact with the aqueous environment, and their long apolar chains promote interactions between them. The hydrophobic chains of each layer face and form a lipophilic compartment around a hydrophilic compartment. Hydrophobic interactions of these lipid bilayers are van der Waals forces, which keep the apolar tails together. Meanwhile, hydrogen bonds and polar interactions between water molecules in the aqueous environment and the polar lipid heads stabilize this organization.

As drug carriers, liposomes are widely used because of their ability to encapsulate hydrophilic, amphiphilic and lipophilic molecules [90]. In addition to surface functionalization methodologies, they can improve pharmacokinetics and the ability to deliver drugs to affected areas [91]. Liposomes also have the advantage of being biodegradable and biocompatible, of low toxicity and the ability for the controlled release of drugs [89,91,92]. Liposomes appear to be a near-perfect system of drug transporters, since their morphology is similar to that of cell membranes and because of their ability to incorporate various substances [93].

The protective phospholipid layer that is usually resistant to pH, body free radicals and enzymatic actions protects ghrelin from degradation until release occurs [89]. Liposomal formulations may also carry ghrelin through the mucosal barrier, protecting ghrelin from metabolism in the nasal cavity [91]. Salade has developed ghrelin liposomes coated with chitosan for the treatment of cachexia by nose-brain delivery [94]. The authors have reported that anionic liposomes were able to protect the drug against the enzymatic degradation of both trypsin (20.6% vs. 0% for ghrelin alone) and carboxylesterase (81.6% vs. 17.2% for ghrelin alone). Ghrelin interacted with the anionic lipid bilayer both by electrostatic and hydrophobic interactions. Their coating with N-(2-hydroxy) propyl-3-trimethyl ammonium chitosan chloride increased 22.9% the capacity for mucin adsorption, with enhanced permeation through Calu3 epithelial monolayers recovering 10.8% of ghrelin in the basal compartment against 0% when using nonloaded ghrelin. Liposomal protection can shield ghrelin from metabolic enzymes in the nasal tissues and promote drug absorption. The dry-powdered form of the developed anionic liposomes coated with chitosan showed stronger adhesion to mucins, higher ghrelin entrapment efficiency, higher enzymatic protection against trypsin and lower ghrelin storage degradation at room temperature [95].

Liposomes have gained attention as promising strategies for the treatment of neurological diseases [87]. The development of central nervous system drug delivery systems is one of the most challenging research topics in the pharmaceutical field. Liposomes improve drug permeation and have the possibility of surface functionalization with different ligands, and their physicochemical characteristics are promising carriers for central nervous system release [91].

Cationic nanostructures are more efficient vehicles for the release of drugs into the central nervous system than the neutral or anionic ones [96]. The electrostatic interaction between negatively charged cell membranes and cationic liposomes increases the liposome uptake by endocytosis adsorption [93].

The addition of polyethylene glycol (PEG) protects the liposome from plasma protein binding, thus forming a protective layer on the surface and avoiding the opsonization process and subsequent elimination of liposomes (Figure 4). Then, PEGylation can prolong the circulation time in the body [97]. The liposomes can improve the entering of other drugs to the central nervous system, and then, it can be used by targeting the receptors expressed on endothelial cells of the brain. The surface of the liposomes can be functionalized with targeting agents that improve the affinity and selectivity of the liposomes for central nervous system administration. Target ligands may be covalently attached on the surface of the liposome or to the ends of the PEG [92,98,99]. Other properties may be included in liposomes for the specific effect of the drug in response to stimuli such as the magnetic field, temperature or changes in pH [91]. Liposomes are currently the type of nanoparticles in most of the studies that have been published for delivery to the brain, thus representing the most advanced material with the most significant potential.

## 6. Conclusions and Future Perspectives

Cachexia is associated with many chronic conditions, infectious diseases and seen in patients after extensive traumatic injuries or sepsis. Cachexia results in a compromised quality of life, increased mortality and greater susceptibility to treatment-related toxicities. With an increasing number of individuals in cachectic conditions associated with the absence of an effective drug intervention, the awareness of cachexia and advances in the development of treatments are urgently needed. We understand that highlighting the practical approaches of the multimodal management of cachexia and its comorbidities is one of the most essentials actions. However, new alternatives for the targeted administration of drugs, therapeutically effective and with biological safety, can reverse the findings of cachectic syndrome completely. The studies considered in this review report is that ghrelin is a promising therapeutic option and may play a role in improving the symptomatic burden of cachexia. Further studies are, however, still needed to overcome the limitations of the administration of exogenous ghrelin and to determine its usefulness in improving the patient’s experience with cachectic syndrome. The use of drug-loaded liposomes to target the central nervous system is still restricted to initial experimental work in cell and animal models or follows preclinical development, with a few now entering clinical trials in humans and requiring further studies. There is still insufficient evidence to support or refute the use of ghrelin in people with cachexia. Further studies are needed to select the appropriate administration route, with a focus on assessing the safety and efficacy. An intelligent drug management system to improve the bioavailability of exogenous ghrelin in people with cachexia can be a suitable alternative to overcome the limitations encountered in the current parenteral administration. Due to biphasic properties, liposomes are the nanostructures most studied for delivering macromolecules like peptides or proteins. Then, liposomes like drug delivery systems have been the proposal to protect ghrelin from physiological instability, increase the retention time and improve its permeation. The naso-cerebral pathway presents itself as a good alternative for ghrelin administration, using a direct connection between the nose and the brain, which can enable its action on the central nervous system. The development of a ghrelin-based cachexia treatment may offer the opportunity to meet the needs of cachectic patients. The naso-cerebral pathway can be one pathway for adherence. However, preclinical developments and clinical trials in humans are needed to establish criteria for the use of liposome-loaded ghrelin and naso-cerebral administration in the treatment of cachexia.

## Figures and Tables

**Figure 1 ijms-21-05974-f001:**
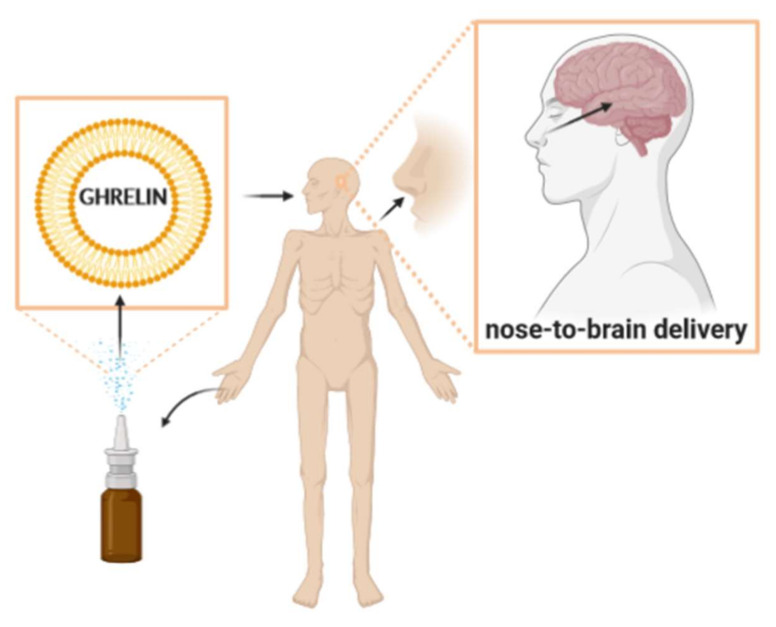
Nose-to-brain delivery of ghrelin-loaded liposomes in cachexia treatments.

**Figure 2 ijms-21-05974-f002:**
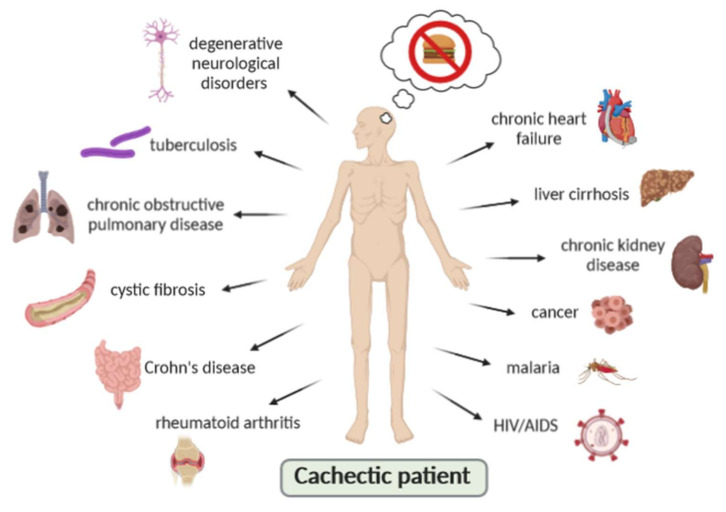
Cachectic patient in association with chronic conditions.

**Figure 3 ijms-21-05974-f003:**
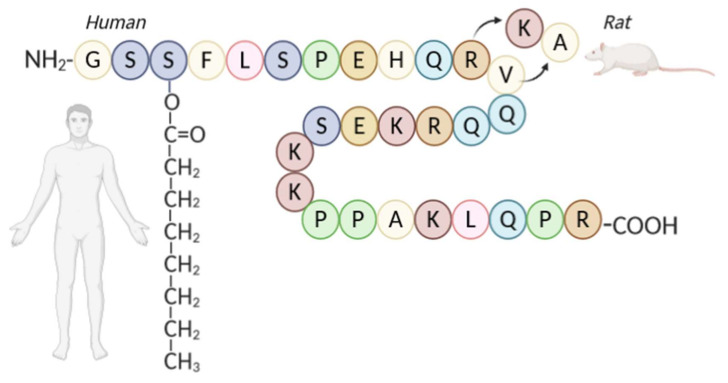
Structure of ghrelin in a human and rat.

**Figure 4 ijms-21-05974-f004:**
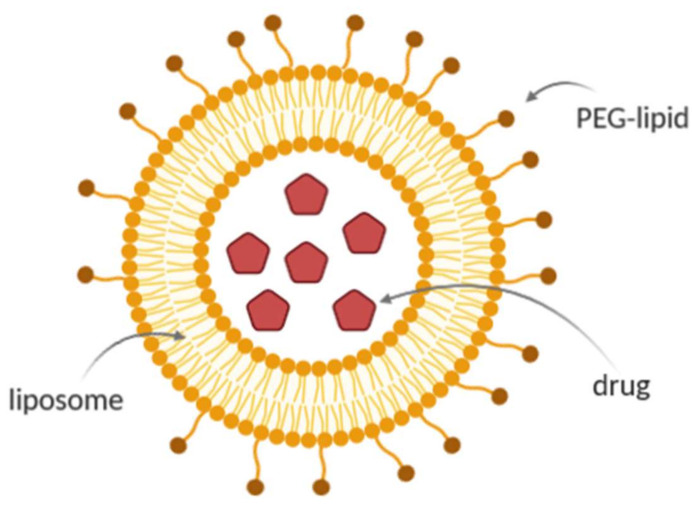
Structure of a liposome with polyethylene glycol (PEG).

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
