# Peer review of "Cachexia: Pathophysiology and Ghrelin Liposomes for Nose-to-Brain Delivery"

_ijms, 2020, doi:10.3390/ijms21175974_

Round 1

Reviewer 1 Report

The manuscript entitled “Cachexia: pathophysiology and ghrelin-liposomes for nose-to-brain delivery” summarizes the literature background of possible treatment of cachexia with ghrelin loaded liposomes. The paper can be improved by addressing the comments shown below such as:

Line 297 describes „Treatment with ghlr has drawbacks”. Please specify the meaning of ghlr, maybe it is only a typo.

Line 365 discusses the limitation of intranasal delivery such as mucociliary clearance. Please complete the section with strategies, which can be used for increasing the residence time of drug on the nasal mucosa e.g. mucoadhesive agents.

The authors discuss only liposomes as suitable carriers for ghrelin. Are there any other nano drug delivery systems, which could be applied for that purpose?

Author Response

The constructive comments and suggestions by the reviewer are really appreciated. We have now completely revised the manuscript. In the following, we respond to the individual remarks (AC = Author’s Comments) and the revised version of the manuscript will be soon transmitted.

AC: Thank you very much for taking the time to read our article.

AC: Regarding your observation on line 297, we corrected the typo and completed the information. Please, look for on lines 295-297: “Treatment with ghrelin has drawbacks, which include a short half-life and the need for infusion or in bolus parenteral administration with multiple side effect events like somnolence, warm feeling, facial warmth, abdominal pain emesis, and vertigo”.

AC: Regarding your observation on line 365; please, look for on lines 360 – 366: “Three routes may contribute independently or synergistically to the transport of drugs, and for the affinity of treatment to a particular pathway, which can be modulated by itself, or by the formulation properties increasing the permeability and decreasing the mucociliary clearance. Between the strategies possible for increase de retention time, systems biomimetics nanoparticulate modulated with an electropositive surface on nanostructure lipid-based is a highly promising approach.”

AC: Regarding your observation on line 468; please, look for on lines 470-473: Due to biphasic properties, the liposomes are the nanostructures most studied for delivering macromolecules like peptides or proteins. Then, liposomes like drug delivery systems have been the proposal to protect ghrelin from physiological instability, increase retention time, and improve its permeation.

AC: The manuscript was carefully revised, pointing out the spelling error. The spell and typos have been carefully checking and revised. The corrections just are highlighted in the red colour.

Reviewer 2 Report

This is an excellent review, very well written and a pleasure to read. It highlights the increase in use of nanoparticulate delivery systems for a different condition than is normal found within the literature and the importance of developing these delivery systems for efficient and efficacious delivery of sensitive drugs.

Author Response

The constructive comments and suggestions by the reviewer are really appreciated. We have now completely revised the manuscript. In the following we respond to the individual remarks (AC = Author’s Comments) and the revised version of the manuscript will be soon transmitted.

AC: Thank you very much for taking the time to read our article.

AC: The manuscript was carefully revised, pointing out the spelling error. The spell and typos have been carefully checking and revised. The corrections just are highlighted in the red colour.